# Surface Modification of PET Fiber with Hybrid Coating and Its Effect on the Properties of PP Composites

**DOI:** 10.3390/polym11101726

**Published:** 2019-10-21

**Authors:** Yapeng Mao, Qiuying Li, Chifei Wu

**Affiliations:** Polymer Processing Laboratory, School of Material Science and Engineering, East China University of Science and Technology, Shanghai 200237, China; maoyp_ecust@163.com

**Keywords:** PET fiber, PP, compatibility, modification

## Abstract

Surface modification fundamentally influences the morphology of polyethylene terephthalate (PET) fibers produced from abandoned polyester textiles and improve the compatibility between the fiber and the matrix. In this study, PET fiber was modified through solution dip-coating using a novel synthesized tetraethyl orthosilicate (TEOS)/KH550/ polypropylene (PP)-g-MAH (MPP) hybrid (TMPP). The PET fiber with TMPP modifier was exposed to the air. SiO_2_ particles would be hydrolyzed from TEOS and become the crystalline cores of MPP. Then, the membrane formed by MPP, SiO_2_ and KH550 covered the surface of the PET fiber. TMPP powder was investigated and characterized by fourier transform infrared spectroscopy, scanning electron microscope (SEM) and thermogravimetric analysis (TGA). TMPP-modified PET fiber was researched by X-ray diffraction and SEM. Furthermore, tensile strength of single fiber was also tested. PET fiber/PP composites were studied through dynamic mechanical analysis and SEM. Flexural properties of composites were also measured. The interfacial properties of PET fiber and PP matrix were indirectly represented by contact angle analysis. Results showed that the addition of TEOS is helpful in homogenizing the distribution of PP-g-MAH. Furthermore, TMPP generates an organic-inorganic ‘armor’ structure on PET fiber, which can make up for the damage areas on the surface of PET fiber and strengthen each single-fiber by 14.4%. Besides, bending strength and modulus of TMPP-modified PET fiber-reinforced PP composite respectively, increase by 10 and 800 MPa. The compatibility between PET fiber and PP was also confirmed to be increased by TMPP. Predictably, this work supplied a new way for PET fiber modification and exploited its potential applications in composites.

## 1. Introduction

Polyethylene terephthalate (PET) fiber is a material with poor wettability and weak interfacial bonding with resin matrix, thus, it is difficult to be used to obtain a high-performance fiber-reinforced resin composite [1]. In particular, the ester groups in PET chains make PET fiber inert toward polypropylene (PP) and polyethylene (PE) matrices, which consist of nonpolar molecular chains in the carbon skeletons [2,3]. In order to improve the compatibility between PET fiber and polyolefin resin matrix to produce a composite with better mechanical strength, and completely embody their advantages including high strength and high modulus, some effective and economic methods to modify the surface of PET fiber are highly desirable. 

In recent years, great attention on polyester fiber has shifted from performance development to fiber recycling. As the most widely used fiber in the world, the amount of PET waste is still growing with sustained and high speed, and the earth’s environment has been seriously damaged. Besides, the accumulation of PET fiber is waste of oil resources. Waste PET could be reused in polymer alloys, toys, soundproof cotton and some other staple fiber products. However, all of these recycling uses only occupy 10% of waste PET, some more effective and economical ways of recycling must be found and industrialized. 

PET fiber-reinforced composites may be a potential method because PET fiber is an organic fiber with high strength and modulus, which would be a substitute of glass fiber. However, how to improve the interfacial properties between PET fiber and the matrix is the most important issue. Thus, the study about the surface modification of fiber has attracted the attention of many research laboratories for decades. To date, extensive research efforts have been devoted to the study of the surface modification of PET fiber. Most of these studies are based on surface chemical etching (alkali treatment), grafting modification, and plasma modification, etc. Through alkali weight reduction processing, the antistatic property, permeability and moisture absorption performance of PET fabric were obviously damaged [4]. Moreover, the fracture strength of PET fiber treated with alkali was also seriously damaged [5]. Plasma processing predominated over the amorphous region of PET fiber. Electron bombardment and free radical reaction were also used for the surface modification of fiber or films [6]. However, previous results [7] showed that the severe plasma treatments in low-pressure plasma led to extensive degradation of the PET fiber, resulting in a decrease of tensile strength. Thus, compared to plasma processing, alkali treatment can be easily controlled by reaction time and temperature. Further, the decrease of fiber strength and weight loss could be designed and predicted [8]. Thus, alkali treatment is often chosen as a basic treatment, while grafting and coating treatment are employed for further functionalized treatment. 

Currently, coating modification of the fiber surface is gradually becoming a hot research topic. The coating on the fiber surface can not only reduce the thermal stress between the fiber and the resin matrix during processing procedure and repair defects on the surface of fiber and damages caused by alkali treatment, but also improve the molecular structure of the fiber surface and improve the wettability of PET fiber in the resin matrix [9]. However, modification of PET fiber through surface coating has rarely been reported. Kwak et al. [10,11,12] modified the surface of PET fiber by thermal treatment with a silver-carbamate complex. First, PET fabric was modified by a series of pretreatments with octaethylene glycol monododecyl, sodium hydroxide, distilled water, ethanol and solution of thiopropyl triethoxy silane in acetone. Then, the treated fabric was immersed in silver 2-ethyl-hexylcarbamate solution and squeezed gently. To form silver nanoparticles, the fabric was transferred into a convection oven at 130 °C for 5 min. After thermal reduction, a continuous layer of silver-coated nanoparticles with sizes between 30 and 100 nm was assembled on the PET fiber. Huang et al. [13] modified PET fiber with titania and PE nanoparticles. Results showed that titania and PE particles were effective in controlling agglomeration of each phase and aided in homogenizing the modified layer. According to these studies, inorganic particles and layers have been proven to be helpful for the dispersion of PET fiber.

Moreover, to retain the original shape and properties of PET fiber during the fabrication of PET fiber-reinforced composites, the melting point of the matrix should be lower than that of PET. PE was selected as a matrix in our previous study, and the results showed that the mechanical properties and thermal stability of PE matrix could be obviously enhanced by PET fiber. Tetraethyl orthosilicate (TEOS) and KH550 have been shown to be successful in coating on fiber with an inorganic layer [14,15]. Owing to the slow hydrolysis reaction in air, TEOS generates uniform SiO_2_ particles on the modified fiber.

Compared to PE, PP is much better in strength and modulus. As a result, PP was set as the matrix to be strengthened. PP-g-MAH (MPP) was selected as a compatibilizer to improve the compatibility between fillers and the PP matrix [16]. 

In this study, SiO_2_ particles were designed to link with the MPP molecular chain through silane coupling agent KH550, which formed an organic-inorganic film by grain structure on the surface of PET fiber obtained from abandoned polyester textiles. This method, taking both advantages of MPP and SiO_2_ particles, is expected to make up for fiber damage caused by alkali treatment and improve the compatibility between the fiber and the PP matrix used in this study. 

Moreover, modified PET fiber and untreated PET fiber-filled PP composites were also prepared for comparative analysis. The effects of different surface modification of PET fiber on the interfacial and mechanical properties between the fiber and the matrix were systematically investigated.

## 2. Experiments

### 2.1. Materials

PET fibers with 20 μm diameter were obtained by laddering from abandoned polyester textiles (commercially available, 75D/24F/22T). These abandoned polyester textiles are leftovers of the daily production in textile mills. The matrix polymer was polypropylene (PP7633) powder with a density of 0.9 g/cm^3^, and MFR = 2 g/10 min (2.16 kg/230 °C, LCY Chemical Corp, Taiwan, China). PP-g-MAH (E516) with a density of 0.93 g/cm^3^ and free maleic anhydride content of less than 1.0% was provided by Ningbo MaterChem Technology Co., LTD (Hangzhou, China). Other agents were of analytical grade and all agents were used as received.

### 2.2. Synthesis of Fiber Modifier

The synthesis of fiber modifier was carried out following the procedure by Chen et al. [17]. Briefly, PP-g-MAH powder was added into boiling xylene solution until it dissolved completely (solution A). TEOS, p-toluenesulfonic acid (PTSA) and xylene were mixed in a mass ratio of 1:0.01:20 (solution B), under stirring for 2 h at room temperature. KH550 and xylene were mixed in a mass ratio of 0.5:20 (solution C). Then, solutions A, B and C were mixed together at 130 °C for 1 h, PET fiber modifier was accomplished, and it was named as TMPP. It is worth noting that the TMPP synthesis process should be carried out without water to avoid the hydrolysis reaction of TEOS. 

### 2.3. Surface Treatment of Polyethylene Terephthalate (PET) Fiber

The PET fiber was immersed in a mixed solution of acetone and water (volume ratio 1:1) for 2 h, and then washed 3 to 4 times with deionized water before being dried for 4 h at 85 °C. Further, the fiber was treated with NaOH solution (0.01 mol L^−1^) at a ratio of 2% (W/V) at 80 °C for 1 h. Then, they were dried and modified with MPP and TMPP by solution dip-coating, respectively. Finally, the modified fiber was dried at room temperature for 3 days. During the course of the formation of TMPP-modified PET fiber, the vapor in the air induced TEOS into a sol–gel reaction, when the modifier was exposed to the air, and SiO2/MPP composite powders were produced on the surface of the PET fiber. A schematic illustration of the interaction between TEOS and KH550 is presented in Figure 1 [18].

### 2.4. Preparation of PET/Polypropylene (PP) Composites

The PET fiber and PP were mixed and homogenized in the mass ratio of 15:85 in an internal mixer with the rotor speed 60 rad s^−1^ at 200 °C for 10 min, and then dried at 100 °C for 4 h in an oven after being smashed. According to the modification of PET fiber, these composites were divided into PP, PP filled with untreated fiber, PP filled with MPP-treated fiber and PP filled with TMPP-treated fiber. 

### 2.5. Characterization

#### 2.5.1. Scanning Electron Microscopy

The morphology of TMPP modifier, the modified PET fiber and the bending sections of PET fiber/PP composites were examined by scanning electron microscopy (SEM, S-3400N, Hitachi, Tokyo, Japan). Samples were treated with spray gold craft before observation.

#### 2.5.2. Fourier Transform Infrared Spectroscopy

Fourier transform infrared (FTIR, Nicolet Magna-IR550, Thermo Electron, New York, USA) spectroscopy was used to identify the modification of the surface of the PET fiber. Samples were loaded on potassium bromide tablets.

#### 2.5.3. Thermogravimetric Analysis

The thermal stability of MPP and TMPP modifiers was tested by thermogravimetric analysis (TGA, Pyris6, Perkin-Elmer, company, Waltham, MA, USA). Samples were heated from ambient temperature to 800 °C at a heat rate of 10 °C/min. The TGA was performed in nitrogen atmosphere with the flow rate of 50 mL/min. 

#### 2.5.4. X-ray Scattering Analysis

The crystallization of the PET fiber was detected by wide angle X-ray scattering (WAXS, PW1830, Philips) at room temperature. Bragg diffraction angles (2θ) were obtained from a diffraction diagram. The inter-planar spacing (d) and crystallite size (t) of the PET fiber were calculated by using Bragg’s law Equation (1) as follows [19]:(1)λ=2dsinθ
where, *d* is the Bragg spacing (Å), *λ* is the radiation wavelength (λ = 1.542 Å) and *2θ* is the diffraction angle (deg).

Then, *t* was calculated by using Equation (2):(2)$$t=\frac{0.9\lambda }{\Delta \omega }cos\theta$$
where, *t* is the crystallite size (Å), and *Δω* is the peak width at half-maximum (rad). The values of *Δω* were determined from diffraction diagrams and converted to radians (radians = degrees × π/180).

#### 2.5.5. Measurements of Interfacial Properties 

The contact angles of substrate relative to water and glycerol were measured using a contact angle meter (OCA20, Dataphysics Company, Esslingen, Germany). Surface energy, interfacial tension and adhesive energy among these phases were calculated by using the Wu’s Equations (3)–(7) as follows:(3)$$\left(1+cos\theta_{H_{2}O}\right)\gamma_{H_{2}O}=4\left(\frac{\gamma_{H_{2}O}^{d}\gamma^{d}}{\gamma_{H_{2}O}^{d}+\gamma^{d}}+\frac{\gamma_{H_{2}O}^{p}\gamma^{p}}{\gamma_{H_{2}O}^{p}+\gamma^{p}}\right)$$
(4)$$\left(1+cos\theta_{C_{3}H_{8}O_{3}}\right)\gamma_{C_{3}H_{8}O_{3}}=4\left(\frac{\gamma_{C_{3}H_{8}O_{3}}^{d}\gamma^{d}}{\gamma_{C_{3}H_{8}O_{3}}^{d}+\gamma^{d}}+\frac{\gamma_{C_{3}H_{8}O_{3}}^{p}\gamma^{p}}{\gamma_{C_{3}H_{8}O_{3}}^{p}+\gamma^{p}}\right)$$
(5)$$\gamma =\gamma^{d}+\gamma^{p}$$
(6)$$\gamma_{12}=\gamma_{1}+\gamma_{2}-4\left(\frac{\gamma_{1}^{d}\gamma_{2}^{d}}{\gamma_{1}^{d}+\gamma_{2}^{d}}+\frac{\gamma_{1}^{p}\gamma_{2}^{p}}{\gamma_{1}^{p}+\gamma_{2}^{p}}\right)$$
(7)$$W_{12}=2{\left(\gamma_{1}^{d}\gamma_{2}^{d}\right)}^{1/2}+2{\left(\gamma_{1}^{p}\gamma_{2}^{p}\right)}^{1/2}$$
where, *θ* is the contact angle, *γ* is the surface energy (mN m^−1^), and *γ*^d^ and *γ*^p^ are the dispersion component and polar component of surface energy, respectively. Subscripts 1 and 2 represent different phases, *γ*_12_ is the interfacial tension (mN m^−1^) between two phases, and *W*_12_ is the adhesive energy of two phases (mN m^−1^).

#### 2.5.6. Mechanical Properties 

The tensile strength of single-fiber was measured by a universal tester (XQ-1, Shanghai) in accordance with GB/T 14337. Bending samples were produced according to standard GB/T 9341-2008 by injection molding (QS-100T, Shanghai Quansheng Plastic Machinery Co., Ltd.). The flexural properties of PP and PET fiber/PP composites were tested using a mechanical property testing machine (WSM-20KN, Changchun).

#### 2.5.7. Dynamic Mechanical Analysis

Dynamic mechanical analysis (DMA) of the composites was carried out using a dynamic mechanical analyzer (Rheogel-E4000, Japan) at a fixed frequency of 11 Hz in bending mode. Results were obtained at a heating rate of 3 °C/min within the temperature range from 70 to 160 °C. A specimen with the dimensions of 40 mm × 5 mm × 2 mm was used in this study.

## 3. Results and Discussion

### 3.1. Modifier Powder

#### 3.1.1. Aggregation Morphology and Size of Modifier Powder

Figure 2 shows the SEM images of MPP and TMPP powders. Figure 2a,b exhibit that the average sizes of MPP and TMPP powders are 7.3 and 2.3 μm, respectively. Compared to MPP, the uniformity of TMPP particle size improves significantly. MPP only dissolve in xylene and only when the temperature is high enough; therefore, after heating is stopped, MPP gradually comes out of the solution along with the decrease of solution temperature. Thus, the instability of the MPP modifier limits its practical application in the industry. After adding TEOS, a colloidal solution was formed by sol–gel reaction and the modifier could remain stable for a long time at room temperature. When exposed to air, the anhydrous condition for TMPP/xylene gel was destroyed, and the hydrolysis process of TEOS with water vapor in the air was triggered; thus, homogeneous TMPP powder was produced. Moreover, MPP and TEOS reacted with silane coupling agent KH550 through anhydride and alkoxy groups separately. The schematic illustration of generation of the product is exhibited in Figure 3. In other words, the TMPP modifier could not only form connections between organic polymer and inorganic SiO_2_ powder, but also improve the solubility of MPP particles in xylene. Therefore, the stability of the TMPP modifier was improved significantly and the aggregation of SiO_2_ particles was prevented, eventually particles acquired the best dispersibility and suitable size. 

#### 3.1.2. Fourier Transform Infrared (FTIR) spectra of MPP and TMPP

FTIR spectra of MPP and TMPP are shown in Figure 4. Peaks at 3411, 2721 and 1627 cm^−1^ are characteristic of amidogens. Only one peak is observed between 3300 and 3500 cm^−1^, and no peak is present around 2000 cm^−1^; thus, the amidogens intended to be secondary amidogens [20]. The peak at 1627 cm^−1^ represents that the secondary amidogens are in close proximity with carbonyl groups. The peak at 1097 cm^−1^ corresponds to the Si–O–bond. Taking the structures of KH550, TEOS, and MPP into consideration, the –NH_2_ in KH550 reacts with maleic anhydride from MPP molecular chains. 

The ideal structures of TMPP modifier and PET fiber are shown in Figure 5. TEOS and KH550 react and form a layer on the surface of the PET fiber, and the sites of –NH_2_ limit the location of MPP molecules. Then, homogenization of MPP distribution is achieved. However, the reaction of KH550 and TEOS could not be controlled, as shown in Figure 5b, the aggregation of SiO_2_ generated a structure as presented in Figure 3, SiO_2_ gathered into centers, and MPP linked with them through the reaction between MAH and –NH_2_.

#### 3.1.3. Thermal Stability of MPP and TMPP Powders

Figure 6 displays the thermogravimetry and differential thermogravimetry analysis (TG and DTG) curves of MPP and TMPP modifiers. Figure 6a exhibits that the initial decomposition temperatures of MPP and TMPP modifiers are 300 and 322 °C, respectively. Figure 6b shows that the DTG peak maxima are at 417 and 437 °C for MPP and TMPP modifiers, respectively. The improvement of thermal stability from MPP to TMPP is attributed to the inhibition of thermal decomposition of TMPP because of the introduction of SiO_2_ granules. Furthermore, the DTG curve of TMPP shows two peaks, one is at about 417 °C and the other is at 437 °C, which indicates the reaction of only part of MPP molecules with SiO_2_. Therefore, the MPP part in the TMPP modifier could be divided into the following two groups: MPP linked with SiO_2_, and pure MPP. Besides, with the increase in the temperature to 800 °C, there is no residual of MPP modifier; however, the percentage of TMPP residual mass is 10 wt.% (Figure 6a), which is supposed to be the content of SiO_2_ generated by TEOS hydrolysis.

### 3.2. Fiber

#### 3.2.1. Crystal Properties and Surface Microstructure of PET Fiber 

Curve (a) in Figure 7, exhibits the existence of only one peak at 17.92°, which is the characteristic peak of PET fiber. Curve (b) shows peaks of MPP-modified PET fiber. Besides the peak 17.86°, the other three peaks represent different crystal faces of MPP. Comparative analysis of curves (b) and (c) indicates that in curve (c), the peak at 25.20° disappears and the area of the peak at 14.24° apparently increases. The characteristic peak of PET at 17.92° is divided into two peaks. Table 1 lists the characteristic diffraction angles obtained from Figure 7, and the calculation results of Bragg spacing and crystallite size. Table 1 summarizes that the crystallite size of PET decreases from 53.63 Å to 53.28 Å and 52.88 Å when it is modified with MPP and TMPP, respectively. TMPP-modified PET fiber exhibited the smallest PET crystal size among all the samples. The results of inter-planar spacing were opposite to those of crystal size, and inter-planar spacing increased after the modification. The results indicate that after alkali treatment, PET chains in crystalline regions are partly broken and form smaller microcrystals [21]. TMPP-modified PET fiber exhibited the smallest crystal size among these samples because effective coating of TMPP on PET fiber promoted the interactions among functional groups present on the surface of PET fiber and TMPP. Peaks at 14.24°, 22.12° and 25.20° are characteristic diffraction peaks of MPP on the surface of the fiber. The extra peak at 18.69° of TMPP-modified PET fiber corresponds to PET linked with SiO_2_ through KH550 [22]. Shifting of the peak of MPP from 22.12° to 21.98° indicates the reaction between MPP and SiO_2_. The reactions between KH550 and TEOS on the surface of TMPP-modified PET fiber lead to the generation of a mesh structure consisting of MPP and Si–O–. Then, the chain segment movement of MPP is restricted, which results in the disappearance of the peak at 25.20°.

These results also show that KH550 and TEOS react and are woven into an armor-like structure on the surface of the PET fiber. They are linked with the Si–O–bond, and the –NH_2_ groups are left and centered on Si–O–circles. The result of FTIR spectroscopy clearly indicated the reaction of MAH of MPP with –NH_2_, which then provides the given space in the armor where MPP is located. Reunification of MPP is prevented and a homogeneous MPP layer is placed on the surface of the PET fiber.

#### 3.2.2. Microstructure of MPP- and TMPP-modified PET Fiber

Figure 8 shows the surface morphologies of untreated and modified PET fiber. A columned and smooth fiber is observed in untreated PET fiber. After being dipped into MPP, the smooth surface was partly made up for by severely agglomerated MPP molecules (Figure 8b). Compared to MPP-modified PET fiber, the surface of TMPP-modified PET fiber is occupied with an integrated compact membrane with embedded and semi-embedded particles (Figure 8c,d), so that the surface roughness and the area of the covered region increased. These results are attributed to the formation of an integrated compact membrane during solvent volatilization and TEOS hydrolysis because of the capillary force when PET fiber soaked with TMPP was exposed to air [23]. The membrane was shaped into a strong cover by SiO_2_ from TEOS and molecular chains of MPP. The TMPP modifier cover restrained the reunion of MPP, and then released more functional groups of MPP than the single-MPP modifier-modified layer. As a result, the TMPP modifier cover was more perfect and stronger than the MPP modifier layer. Moreover, the organic–inorganic ‘armor’ on the PET fiber led to the dramatic increase in the roughness of the PET fiber. The specific surface area of the fiber was enlarged, leading to the generation of the molecular tangles between the fiber and the matrix [24]. All these changes led to the increase in the interfacial bonding strength between the fiber and the matrix and were found to be beneficial for the ultimate performance of PET fiber-reinforced composites.

#### 3.2.3. Tensile Test for Single Fiber

Tensile strength of single fiber is an important index to measure the mechanical properties of fiber. Figure 9 shows that the values of tensile strength of untreated PET fiber, MPP-modified PET fiber, and TMPP-modified PET fiber are 1.25, 1.37 and 1.41 GPa, respectively. MPP and TMPP modifiers led to the increase in the tensile strength of PET fiber by 10.4% and 14.4%, respectively. It is well known that PET fiber is composed of highly oriented PET molecular chains. However, numerous terminal hydroxyl groups were generated from the breaking of PET molecular chains on the fiber surface after alkali treatment. The terminal hydroxyls reacted with the epoxy groups from MPP, and MPP was linked to the surface of PET fiber via –C–O–linkages. Therefore, MPP could apparently increase tensile strength of single fiber. Besides the effect of the single-MPP modifier, the TMPP modifier could form a special structure with the MPP-SiO_2_ link, through TEOS, MPP, and KH550. The special structure covered the fiber like a crust, and then the coating was tighter and more homogeneous. The structure also made up the hollows on the fiber surface, optimized the surface structure, improved stress transfer ability, and finally, enhanced the strength and carrying capacity of the fiber [25]. TMPP modification could strengthen the PET fiber and improve the compatibility between the fiber and the matrix at the same time.

### 3.3. Composites

#### 3.3.1. Interfacial Properties

To verify the effect of MPP and TMPP modifiers on surface modification, interfacial properties of the composites were analyzed by using contact angle. Surface energy, interfacial tension and adhesive energy were calculated and are listed in Table 2 and Table 3. Table 2 summarizes that the surface energies of PP, MPP, and TMPP modifier are 22.26, 15.59 and 22.75 mN m^−1^, respectively. Clearly, the surface energy of PP was closer to that of the TMPP modifier than the MPP modifier, which indicates that the TMPP modifier has better compatibility with PP than with the MPP modifier. It is due to the fact that MPP is mostly evenly coated on the surface of the PET fiber when modified with the TMPP modifier, which leads to the surface polarity of the TMPP-modified PET fiber being more similar to that of PP than the MPP-modified PET fiber. Table 3 presents that the interfacial tension of MPP/PP is higher than that of TMPP/PP, and the adhesive energy of MPP/PP is lower than that of TMPP/PP. According to He’s work [26], smaller interfacial tension causes tighter bonds between two phases. As a result, the addition of TEOS and KH550 in the TMPP modifier regulates the location and direction of MPP. In a word, the modification of the TMPP modifier was better than that of the MPP modifier.

#### 3.3.2. Interface Compatibility of Composite Materials

Figure 10 shows the DMA curves, exhibiting the dynamic properties of PET fiber/PP composites reinforced with different modified PET fibers. Compared to PP, the storage modulus of PET fiber/PP composites increased with the addition of PET fiber, as shown in Figure 10a. The cooperation of modifier and PET fiber leads to the obvious increase in the storage modulus of the PP composite. This is attributed to the fact that compared to untreated PET fiber, the MPP-modified PET fiber exhibits larger specific surface area and better compatibility with PP, which results in higher storage modulus. The TMPP-modified PET fiber leads to the further increase in the storage modulus of the PP composite, because of a more homogeneous modifier layer, larger specific surface area, and higher single-fiber tensile strength than the MPP-modified PET fiber. Figure 10b shows the loss modulus curves of PP and PP composites. Results indicate that the glass-transition temperature ($T_{g}$) of composites increased when filling with untreated PET fiber and MPP-modified ones. This is attributed to the fact that after modification, the interfacial compatibility of PET fiber and PP gets promoted, and the interaction force increases; thus, the energy required for releasing molecular chains also increases. As a result, the loss modulus and $T_{g}$ of composites increase [27,28]. Moreover, compared to the MPP-modified PET fiber-reinforced PP composite, the $T_{g}$ of the TMPP-modified PET fiber-reinforced PP composite decreases. When PET fiber is modified with the TMPP modifier, the addition of TEOS imports a few SiO_2_ particles at the interface between fiber and the matrix, which works as a nucleator for PP. Crystallinity of PP increases for the SiO_2_, and the amorphous phase decreases. As a result, in contrast with the PP composite filled with the MPP-modified fiber, the $T_{g}$ of the PP composite filled with the TMPP-modified PET fiber decreases.

The bending properties of PP and PET fiber/PP composites are shown in Figure 11. Compared to PP, the bending strength and modulus of PET fiber-reinforced composites improved significantly. The properties of the TMPP-modified PET fiber-reinforced PP composite was the best among all the samples and increased by 10 and 800 MPa more than the PP composite filled with untreated PET fiber, with respect to bending strength and modulus respectively, which resulted from the fact that the coatings on PET fiber led to the improvement in the filler-matrix compatibility. In particular, the PET fiber is evenly coated with a layer of membrane in the TPMM-modified PET fiber, similar to a shell to provide the resistance to stress deformation. 

Figure 12 exhibits SEM images, showing the bending sections of different PET fiber/PP composites. Figure 12a demonstrates that the untreated PET fiber apparently separates from the matrix. The PET fibers still remain smooth and unbroken when they are pulled out from the PP matrix. Different from the untreated PET fiber-filled PP, the number of pulled-out PET fiber decreases, as shown in Figure 12b, and the PET fiber is coated with a layer in MPP-modified PET fiber-filled PP. This is because the fiber is partly coated with non-polar groups in the MPP-modified PET fiber and the interfacial compatibility between the fiber and the PP matrix is improved. Figure 12c exhibits the cross-section of TMPP-modified PET fiber-filled PP composites, showing that the PET fiber is finely dispersed in the matrix and is inlaid into the matrix. Moreover, the fiber and the matrix were combined as a whole through a homogeneous amphiprotic TMPP membrane. In general, the TMPP modifier improved the interfacial compatibility of the composites and optimized the dispersion of fiber in the matrix. As a result, the PP composite filled with the TMPP-modified PET fiber presented the best bending properties. 

## 4. Conclusions

In summary, the TMPP modifier was firstly obtained in an anhydrous atmosphere, and then a uniform and homogeneous TMPP membrane was synthesized by sol–gel method on the surface of the PET fiber. Finally, a TMPP-modified PET fiber-reinforced PP composite was also produced. FTIR, XRD, and SEM analyses confirmed the formation of a membrane with chemical construction and micro-topography on the PET fiber surface after modification. The TMPP membrane was formed by amid bonds, which came from the reaction between KH550 and MPP. TGA showed the presence of SiO_2_ particles and indicated that only partial MPP participated in the reaction with KH550 on the fiber surface. The TMPP membrane increased single-fiber strength by 14.4% and improved the bending strength and modulus of the PET fiber/PP composite by 21% and 34% respectively, compared to the untreated fiber-filled composite. The analysis of contact angles and DMA testified to the increase of compatibility between the TMPP-modified fiber and the PP matrix. The coating modification effectively and conveniently improved the interfacial properties between the fiber and the matrix as well as the mechanical properties of the PET fiber. This method is expected to be an important direction for further research on fiber surface modification and develops the applications of recycled PET fiber in composites.

## Figures and Tables

**Figure 1 polymers-11-01726-f001:**
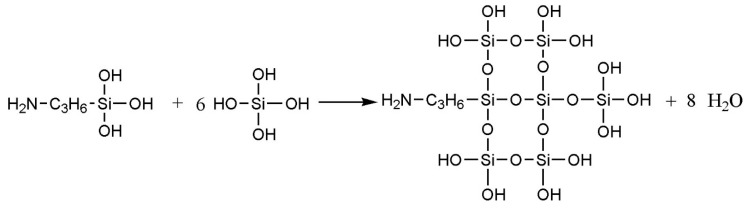
Interaction between TEOS and KH550.

**Figure 2 polymers-11-01726-f002:**
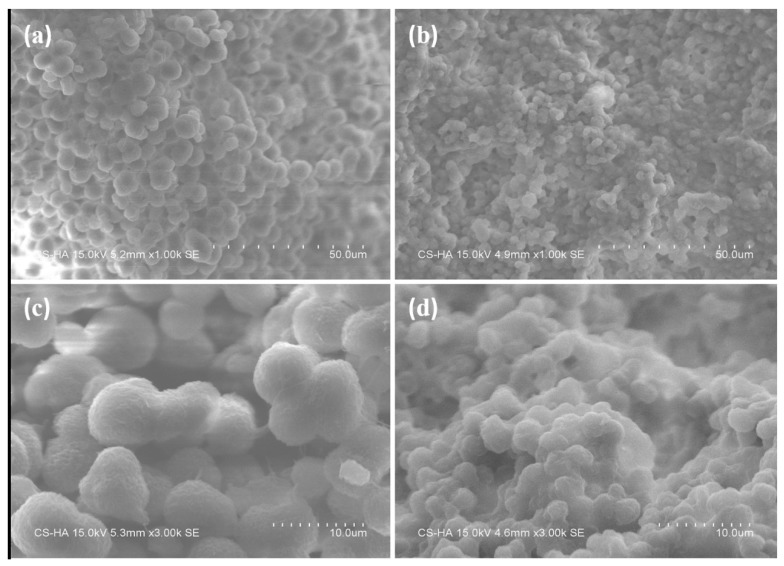
Scanning electron microscope (SEM) images of MPP and TMPP modifiers after hydrolysis. ((**a**) MPP powder, 1000 times, (**b**) TMPP powder, 1000 times, (**c**) MPP powders, 3000 times, (**d**) TMPP powder, 3000 times).

**Figure 3 polymers-11-01726-f003:**
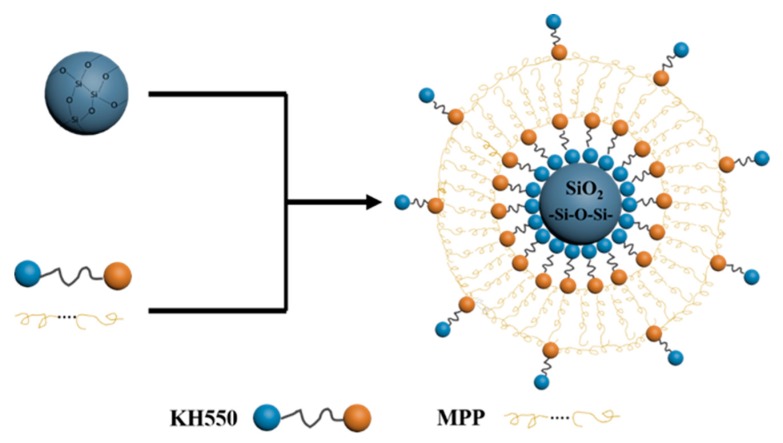
The structural representation of SiO_2_/MPP modifier.

**Figure 4 polymers-11-01726-f004:**
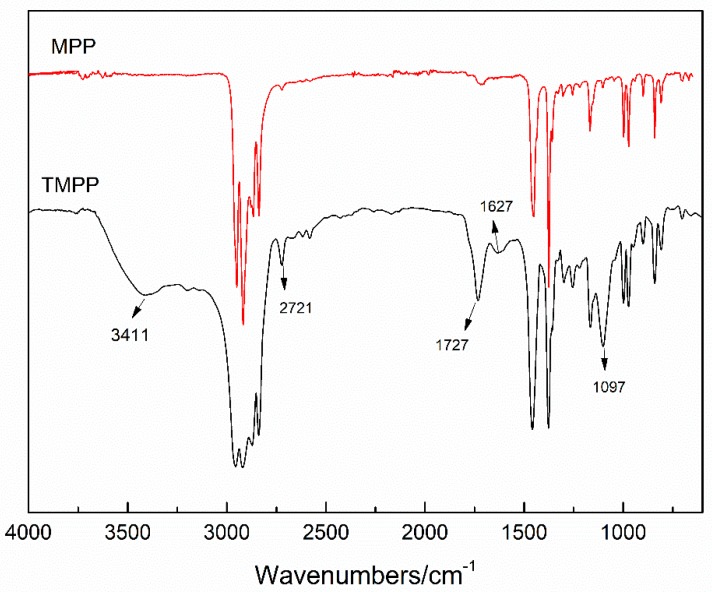
Fourier transform infrared (FTIR) spectra of MPP and TMPP modifiers.

**Figure 5 polymers-11-01726-f005:**
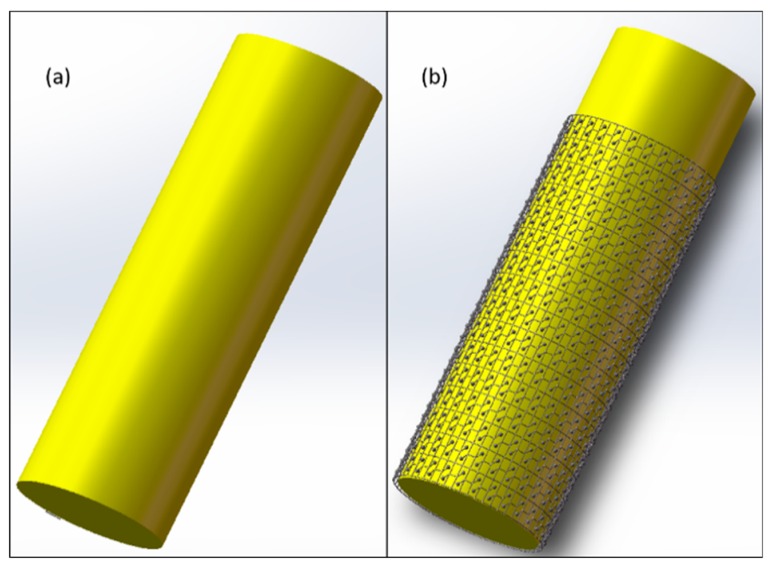
Ideal model of (**a**) PET fiber and (**b**) TMPP-modified PET fiber.

**Figure 6 polymers-11-01726-f006:**
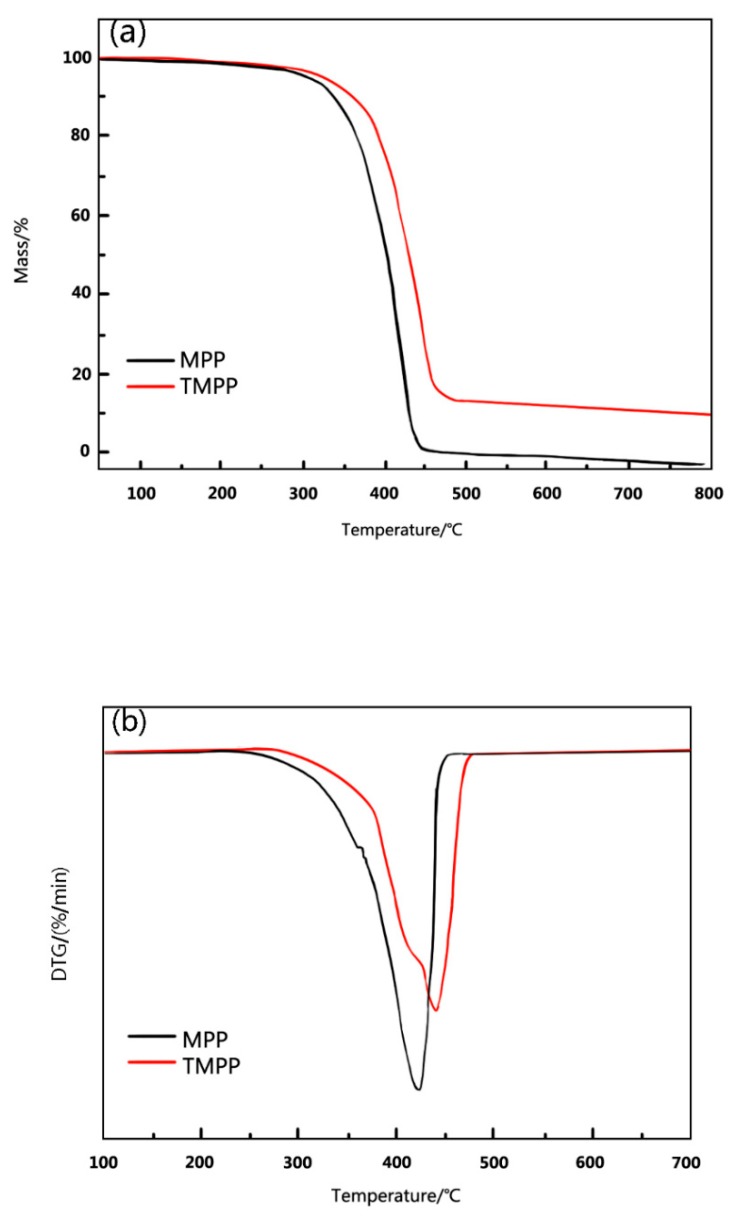
The (**a**) TG and (**b**) DTG curves of MPP and TMPP modifiers.

**Figure 7 polymers-11-01726-f007:**
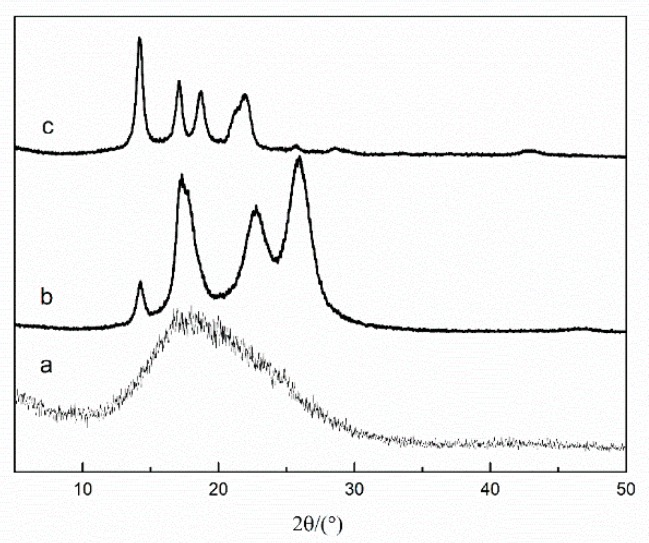
Curves of (**a**) PET fiber, (**b**) MPP-modified PET fiber and (**c**) TMPP-modified PET fiber.

**Figure 8 polymers-11-01726-f008:**
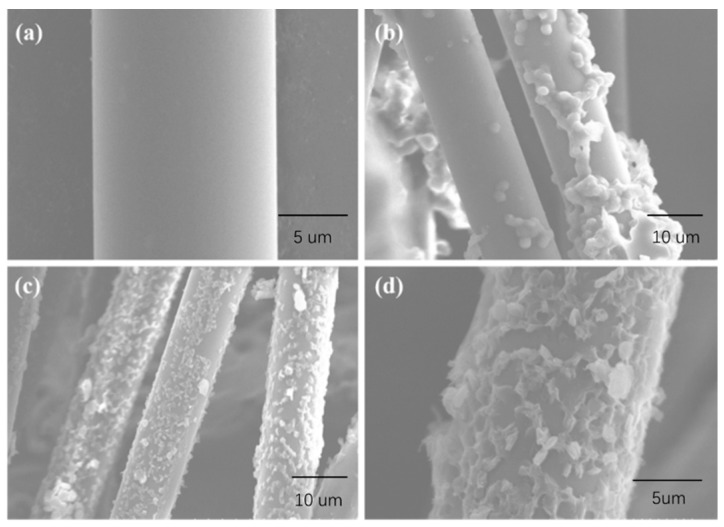
Microstructure of (**a**) untreated PET fiber, (**b**) MPP-modified PET fiber, and (**c**) and (**d**) TMPP-modified PET fiber.

**Figure 9 polymers-11-01726-f009:**
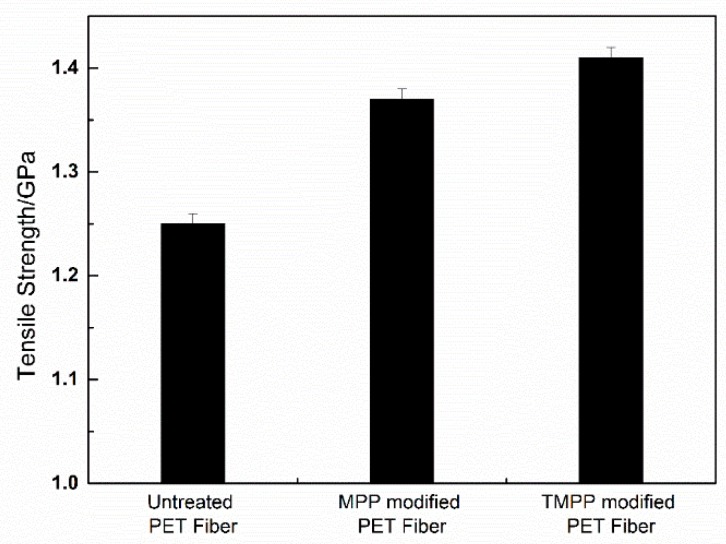
Tensile strength of single fiber.

**Figure 10 polymers-11-01726-f010:**
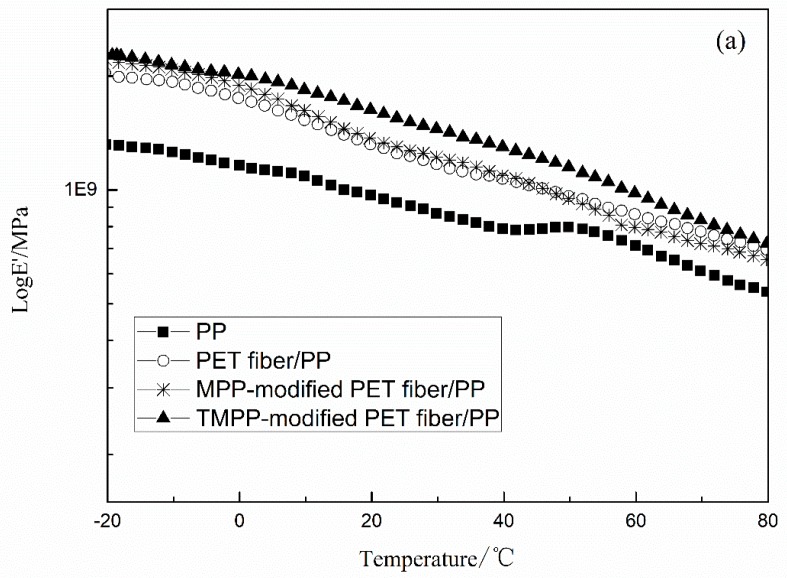
Dynamic properties of different PP composites (**a**): storage modulus, (**b**): loss modulus.

**Figure 11 polymers-11-01726-f011:**
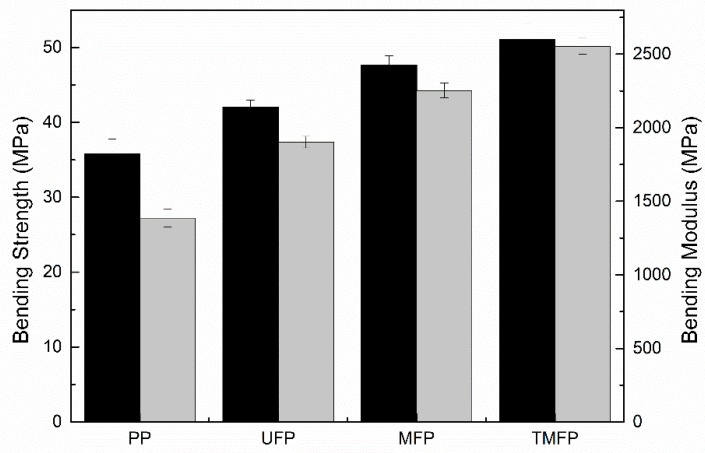
Bending properties of different PP composites ((**a**) PP, (**b**) PP composite filled with untreated PET fiber, (**c**) PP composite filled with the MPP-modified PET fiber, (**d**) PP composite filled with the TMPP-modified PET fiber).

**Figure 12 polymers-11-01726-f012:**
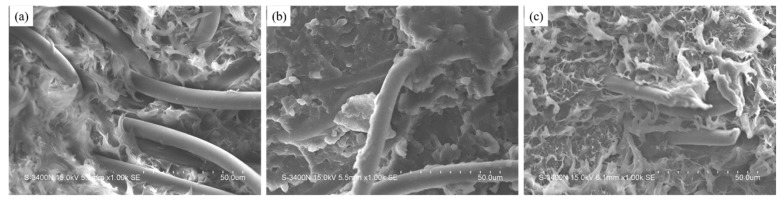
SEM images of PP composites filled with (**a**) untreated PET fiber, (**b**) MPP-modified PET fiber and (**c**) TMPP-modified PET fiber.

**Table 1 polymers-11-01726-t001:** Diffraction angles (2θ), inter-planar spacing (d) and crystallite size (t) of PET fiber, and modified PET fiber.

Sample Code	Peak NO.	2θ(^o^)	d = λ/(2sinθ) (À)	t = 0.9λ/Δωcosθ (À)
PET fiber	1	17.92	4.95	53.63
MPP-modified PET fiber	1	14.24	6.22	54.26
	2	17.86	4.97	53.28
	3	22.12	4.02	51.66
	4	25.20	3.53	62.77
TMPP-modified PET fiber	1	14.24	6.22	54.35
	2	17.18	5.16	52.88
	3	18.69	4.75	52.16
	4	21.98	4.04	51.24

**Table 2 polymers-11-01726-t002:** Surface energies between the MPP modifier, the TMPP modifier and the PP matrix.

Sample	Contact Angle *θ*H_2_O (^o^)	Contact Angle *θ*C_3_H_8_O_3_(^o^)	Surface Energy γ (mN/m)	Dispersion Component *γ*^d^ (mN/m)	Polar Component *γ*^p^ (mN/m)
MPP	107.49	97.99	15.59	6.50	9.09
TMPP	97.99	81.99	22.75	14.86	7.89
PP	97.50	85.74	22.26	13.49	8.77
H_2_O	-	-	72.80	21.80	51.00
C_3_H_8_O_3_	-	-	63.40	37.00	26.40

**Table 3 polymers-11-01726-t003:** Interfacial tensions and adhesive energies between the MPP modifier, the TMPP modifier and the PP matrix.

Sample	Interfacial Tension $\gamma_{12}\;(\mathrm{mN}/m)$	Adhesive Energy $W_{12}\;(\mathrm{mN}/m)$
MPP/PP	2.45	36.59
TMPP/PP	0.11	44.95

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
