# Peer review of "Surface Modification of PET Fiber with Hybrid Coating and Its Effect on the Properties of PP Composites"

_polymers, 2019, doi:10.3390/polym11101726_

Round 1

Reviewer 1 Report

The review article (Manuscript Number polymers-614634) presents information about the modification of polyethylene terephthalate (PET) fibers through solution dip-coating using a tetraethyl orthosilicate (TEOS)/KH550/PP-g-MAH (TMPP) hybrid. The authors stated that the addition of TEOS is helpful in homogenizing the distribution of PP-g-MAH. What is important, TMPP generates an organic-inorganic ‘armor’ structure on PET fiber, which can make up for the damaged areas on the surface of PET fiber and strengthen each single-fiber by 14.4%.
On the whole, the manuscript is fairly well-written and logically arranged. The overall originality of the review article concept used here is medium-high. Nevertheless, I would recommend publication of this review article in Polymers on the condition a minor revision of the manuscript will be carried out and the following points will be taken into consideration.

Detailed comments:

1. The abstract needs to be well written with future prospects of the work and describe in short the concept of a modification of PET fibers.
2. A more detailed advantage of the present field must be mentioned in the Introduction. Furthermore, an introduction should be worked out - so as to show the full state of knowledge on PET fibers. Extension to look at these issues and also provide other techniques for receiving them should also be provided.
3. The conclusion reflects an overall summary of the field with further extension and includes future prospective - I would suggest clarifying this section.
4. The chapter appears to be a collection of data from other research papers, however, the authors' self-opinion is of importance while drafting a chapter of this type.

After completing the above-mentioned corrections this work will be more readable. Therefore, it will be useful for the readers of the Polymers.

Author Response

Point 1: The abstract needs to be well written with future prospects of the work and describe in short the concept of a modification of PET fibers. 

Response 1: Thanks for your suggestion. These sentences have been added in abstract:

PET fiber with TMPP modifier was exposed to the air, SiO2 particles would be hydrolyzed from TEOS and become the crystalline cores of MPP. Then, the membrane formed by MPP, SiO2 and KH550 covered the surface of PET fiber. Predictably, this work supplied a new way for PET fiber modification and exploited its potential applications in composites.

Point 2: A more detailed advantage of the present field must be mentioned in the Introduction. Furthermore, an introduction should be worked out - so as to show the full state of knowledge on PET fibers. Extension to look at these issues and also provide other techniques for receiving them should also be provided.

Response 2: Thanks a lot for this advice. These sentences have been added in the introduction:

In recent years, great attention on polyester fiber has shifted from performance development to fiber recycling. As the most widely used fiber in the world, the amount of PET waste is still growing with sustained and high speed, and the earth’s environment has been seriously damaged. Besides, the accumulation of PET fiber is waste of oil resources. Waste PET could be reused in polymer alloys, toys, soundproof cotton and some other staple fiber products. However, all these recycle uses only occupies 10% of waste PET, some more effective and economical recycle ways must be found and industrialized. PET fiber reinforced composites maybe a potential method because PET fiber is an organic fiber with high strength and modulus, which would be a substitute of glass fiber. However, how to improve the interfacial properties between PET fiber and the matrix is the most important issue. Thus, the study about the surface modification of fiber has attracted the attention of many research laboratories for decades.

Point 3: The conclusion reflects an overall summary of the field with further extension and includes future prospective - I would suggest clarifying this section.

Response 3: Thanks a lot for this advice. The conclusion has been rewritten as follows:

In summary, TMPP modifier was firstly obtained in an anhydrous atmosphere, and then a uniform and homogeneous TMPP membrane was synthesized by sol–gel method on the surface of PET fiber. Finally, TMPP-modified PET fiber reinforced PP composite was also produced. TMPP membrane was formed by amid bonds, which came from the reaction between KH550 and MPP. This method is expected to be an important direction for further research on fiber surface modification and develops the applications of recycled PET fiber in composites.

Point 4: The chapter appears to be a collection of data from other research papers, however, the authors' self-opinion is of importance while drafting a chapter of this type.

Response 4: Thanks a lot for this advice.

In the present study, the surface modification of PET fiber was on the base of reference [17]. However, I have transferred the hybrid from ‘SiO2/SMPU’ to ‘SiO2/MPP’. Phosphoric acid, acetone, and absolute ethanol and N,N-dimethylacetamide were replaced by p-toluenesulfonic acid, xylene, and PP-g-MAH.

In addition, self-opinions have been given in every chapter according to the results.

Reviewer 2 Report

The manuscript and the work presented is interesting and well written. However, it has to undergo minor revision before publication. Images of higher resolution should be provided. References are considered up-to-date utmost. Results and methodology are clear and concise, while conclusion are clearly drawn from results. Additionally, in Figure 8, the magnification should be noted. Bending properties should include error bars.

Author Response

Response to Reviewer 2 Comments

Point 1: In Figure 8, the magnification should be noted.

Response 1: Thanks for your suggestion. The graph has been replaced as the attachment.

Point 2: Bending properties should include error bars.

Response 2: Thanks a lot for this advice. The graph has been replaced as the attachment.

Point 3: References are considered up-to-date utmost

Response 2: Thanks a lot for this advice. Reference [9] has been replaced by ‘Fang, Y.; Liu, X.; Tao, X. Intumescent flame retardant and anti-dripping of PET fabrics through layer-by-layer assembly of chitosan and ammonium polyphosphate. Prog. Org. Coatings 2019, 134, 162-168.
